# Application progress and clinical translation of artificial intelligence-assisted endoscopic diagnosis of early esophageal squamous cell carcinoma

## Abstract

Early and accurate diagnosis of esophageal squamous cell carcinoma (ESCC) is key to improving patient outcomes. Endoscopy plays a central role in its early diagnosis and treatment. Despite advances in imaging technology, clinical challenges remain, including missed diagnosis of flat lesions and subjective variability among physicians. Artificial intelligence has demonstrated transformative potential in endoscopic diagnosis, offering an effective solution to overcome existing bottlenecks and improve the accuracy of early ESCC identification. This article systematically reviews the clinical value of artificial intelligence in ESCC detection, classification, and invasion depth prediction, aiming to identify the technical advantages, bottlenecks in clinical translation, and future research directions.

## 1 Introduction

Esophageal squamous cell carcinoma (ESCC) accounts for 85% of the global esophageal cancer incidence. Early diagnosis offers a 95% five-year survival rate. However, traditional endoscopy relies on physician experience, resulting in a 7.3%-9.4% miss rate for early lesions. Artificial intelligence offers a new paradigm for improving early ESCC diagnosis by analyzing endoscopic image features using deep learning convolutional neural networks. This article reviews the clinical value of artificial intelligence in ESCC detection, classification, and invasion depth prediction, aiming to identify the technical advantages, bottlenecks in clinical translation, and future research directions.

## 2 Deep learning-assisted lesion detection

### 2.1 Single modality

#### 2.1.1 WLI mode

Timely detection of esophageal squamous cell carcinoma (ESCC) is crucial to improving patient prognosis. White light endoscopy (WLI), as the main means of clinical screening, faces challenges such as insufficient sensitivity and specificity (sensitivity as low as 62%) and dependence on physician experience for diagnosis especially in resource-limited areas[1–4]. To break through this bottleneck, many studies have focused on developing artificial intelligence systems based on deep convolutional neural networks (DCNNs) to improve the detection and localization efficiency of ESCC under WLI. The multicenter DCNN model developed by Liu et al. achieved a sensitivity of 92.6% and an accuracy of 85.7% in internal validation, and maintained a sensitivity of 89.5% and an accuracy of 84.5% in external validation. Its accuracy of lesion boundary delineation (93.4%) was significantly better than that of senior physicians (78.6%), and the processing time of a single image was only 17 milliseconds, which was more than 5000 times higher than manual efficiency. This system filled

the technical gap of real-time boundary delineation for WLI for the first time[1]. The progressive multi-granularity DCNN model constructed by Tang et al. achieved a sensitivity of 97.9% and a specificity of 88.6% (AUC 0.954) in the internal validation set, and maintained an AUC of more than 0.925 in the external validation set. Its outstanding advantage is that it can distinguish ESCC from reflux esophagitis/normal mucosa, and still maintains 100% sensitivity in low-quality images. After the model is assisted, the sensitivity of junior endoscopists increased by 21.1%[2]. Cai et al. verified the clinical auxiliary value of their deep neural network computer-assisted detection system. The system sensitivity and accuracy reached 97.8% and 91.4% respectively, which were significantly higher than those of the junior physician group (77.2%). After assistance, the sensitivity of physicians at all levels increased by 15%, especially the accuracy of the junior group improved by 11.6%, highlighting the role of AI in shortening the experience gap[3]. Feng et al. further broke through the limitation of equipment dependence and developed the first universal AI system compatible with Olympus and Fujifilm dual platforms. The model, based on the bilinear pooling attention network, achieved 96.64% sensitivity and 95.35% specificity in internal verification, and maintained 91.60% sensitivity in external verification. Its lesion heat map localization assistance increased the specificity of junior physicians by 39.34% in external verification[4]. These studies have jointly verified the core advantages of AI models in detecting ESCC under WLI: the sensitivity (89.5%-97.8%) and accuracy (84.5%-91.4%) consistently surpass those of non-expert physicians, especially in terms of boundary delineation, which is about 15 percentage points higher than manual delineation[1, 4]. At the same time, all models achieve millisecond-level image processing speed and have the potential for real-time application; however, the performance differences between different models are reflected in edge tasks. For example, the Liu model performs outstandingly in boundary intersection over union (mIoU 70.3%)[1], while the Feng model demonstrates cross-platform generalization capabilities[4]. The main reasons for the differences lie in the model architecture (such as whether the boundary segmentation module is integrated) and the diversity of training data (such as whether images of multiple brands of equipment are included). Comprehensive analysis shows that AI systems promote clinical practice in two ways: first, they directly provide high-precision diagnosis (with sensitivity comparable to that of expert physicians); second, they serve as auxiliary tools to significantly improve the diagnostic level of physicians at all levels (with an average sensitivity increase of 15%-21.1%)[1–4]. However, the current researches are mainly concentrated on static image verification, lack of prospective research on dynamic video. In the future, it is necessary to integrate multimodal data and promote hardware integration to achieve large-scale implementation[1, 2, 4].

### 2.1.2   NBI mode

Magnifying endoscopy with narrow band imaging (ME-NBI) is the gold standard for the diagnosis of early-stage ESCC. It visualizes the morphological changes of intrapapillary capillary loops (IPCLs) in the mucosal surface, enabling microvascular classification (Japan Endoscopic Society classification, types A/B1/B2/B3). However, IPCL classification is highly dependent on endoscopist experience, with significant interobserver variability (interobserver diagnostic agreement kappa values are only 0.40-0.60). AI is urgently needed to improve diagnostic standardization. The double-labeling fully convolutional network developed by Zhao et al. first achieved pixel-level segmentation and classification of IPCLs, achieving a lesion-level accuracy of 89.2% in 1383 esophageal lesions, close to the level of experts with more than 15 years of experience (92.0%), and significantly improved the misdiagnosis problem of B1/B2 (sensitivity 87.6%/93.9%)[5]. Uema et al. further optimized microvascular classification by pre-training the ResNeXt-101 model. Their computer-aided diagnosis (CAD) system achieved an overall accuracy of 84.2% on 747 ME-NBI images covering B3 vessels, especially improving the diagnostic accuracy of B2 (67% vs. 63.6% for physicians), and used Grad-CAM++ visualization to verify the decision-making focus on the vascular area[6].

Multicenter studies have enhanced the generalization ability of AI models: the system developed by Yuan et al. based on the HRNet+OCR architecture achieved an accuracy of 89.8%-91.3% in the IPCL subtype classification in a cross-institutional validation set (7094 ME-NBI images), assisted junior physicians to improve the diagnostic accuracy by 6.5% (84.7% vs. 78.2%), and improved the observer consistency (Kappa value increased to 0.545)[7]. Everson et al. constructed the first clinically interpretable convolutional neural network (CNN), which visualized the abnormal area of IPCL in real time through class activation maps (CAMs), and achieved an accuracy of 91.7% (AUC 95.8%) on 67,742 ME-NBI images, which was comparable to the performance of the European, American and Asian expert group (F1 score of Asian experts was 98%)[8].

The difference in model performance stems from the architecture design and training strategy: early studies used basic segmentation networks (such as fully convolutional network) to improve pixel-level accuracy, while recent work introduced pre-trained models (ResNeXt, HRNet) and interpretable modules to optimize classification robustness[5–8]. Data scale and quality also affect generalization. Cross-center validation (three hospitals) and B3 type sample enhancement (such as Uema and other integrated learning strategies) effectively alleviate model bias[6, 7]. Performance comparison shows that B2 type blood vessels become a diagnostic bottleneck due to their large morphological variation (physician accuracy rate of 63.6-67%), while the AI system significantly improves its recognition ability through quantitative feature extraction (accuracy rate of 67-85.7%)[5–7].

Overall, the AI diagnostic system for single-modality ME-NBI has achieved near-expert-level IPCL classification performance. Real-time visualization aids (CAMs and ROI annotation) improve diagnostic consistency among junior physicians and address the core difficulty of subjective variability in the interpretation of type B2 vessels. Future efforts require lightweight design to adapt to primary care devices and integrate video stream analysis for dynamic lesion assessment.

## 2.2 Bimodal or multimodal mode

Endoscopic diagnosis of early ESCC is highly dependent on physician experience. Conventional WLI has a missed diagnosis rate of up to 6.4%. Single imaging modality has significant limitations, so AI systems integrating multiple endoscopic modalities have become a research hotspot. Yuan et al. developed an AI system based on a DCNN that simultaneously integrates four endoscopic modes: WLI, non-ME NBI, iodine staining, and ME NBI. The system was trained and validated on 53,933 images and 142 videos from five centers. The system demonstrated excellent performance in both internal and external validation sets (sensitivity 92.5–99.7%, specificity 78.5–89.0%, and AUC 0.906–0.989), comparable to that of 11 experienced endoscopists overall. Furthermore, the system demonstrated significantly higher sensitivity for epithelial-confined ESCC in the WLI mode (90.8% vs 82.5%) than the endoscopists[9]. The system achieved real-time video processing (60 frames per second) and could handle common clinical interferences such as mucus and bubbles[9]. In subsequent studies, the team further optimized the algorithm and developed a new system based on YOLACT++. Under real-time multimodal endoscopic imaging, it not only detected tiny (about 3 mm) flat ESCC, but also achieved accurate delineation of the lesion boundary. The system can also be directly integrated into the endoscopic device[10].

In addition to integrating more modes, different teams have explored the effectiveness of dual-modality systems. The system developed by Guo et al. is specifically for the non-magnified and magnified modes of NBI. It uses the SegNet architecture to generate real-time probability heat maps (high-risk lesions are marked in yellow). The sensitivity of each lesion in video verification reached 100%, although the sensitivity of each frame of non-magnified video (60.8%) was lower than that of magnified video (96.1%), mainly because motion blur affects feature extraction[11]. Meng et al. compared the performance differences between WLI and NBI dual modes. The CAD system based on the improved YOLO v5 algorithm achieved an AUC of 0.982 on an independent test set. The accuracy of the NBI mode (94.6%) was significantly higher than that of the WLI mode (89.5%). In particular, flat lesions (Paris type 0-IIb) were more likely to be misjudged under WLI [12]. The system improved the diagnostic capabilities of non-expert physicians, and the accuracy of non-experts increased from 78.3% to 88.2% after reference to CAD[12].

The performance differences between different studies are closely related to the number of integrated modalities, task complexity and algorithm selection. Multimodal systems (such as the Yuan study that integrated four modalities[9, 10]) optimized the overall performance through complementary imaging features, especially in the detection of early cancer under WLI[9]. In contrast, dual-modal systems performed well in specific scenarios: NBI-specific systems achieved the highest sensitivity per lesion (100%) for typical lesions but depended on image quality[11], while WLI/NBI dual-modal systems improved the recognition stability of flat lesions through algorithm improvements (such as optimizing loss functions) [12]. It is worth noting that all systems confirmed the auxiliary value of AI for non-expert physicians and can narrow the gap in diagnostic experience[9, 12]. Future directions need to focus on real-time multimodal interaction (such as simultaneous display of AI results of multiple imaging), algorithm generalization improvement (covering rare lesions) and prospective clinical verification to promote the evolution of AI-assisted endoscopy from static image analysis to dynamic multimodal integrated decision-making[9–12].

## 2.3  Real-time detection

Currently, the development of real-time video diagnostic systems based on AI has become a research hotspot. Representative studies have used CNNs to achieve dynamic analysis of endoscopic videos. For example, the AI system developed by a Japanese team used 23,977 endoscopic images (WLI and NBI/blue laser imaging (BLI)) to train the model, which can distinguish superficial cancer (EP-SM1) from deep invasive cancer (SM2-3) in real time. In 102 independent video verifications, the AI had a specificity of 98.7% under non-ME and a sensitivity of 71% under ME (better than the expert group's 42%)[13]. Similarly, the system developed by the Chinese team combined the SegNet architecture and achieved dual-mode real-time diagnosis after training on 6473 NBI images. In the video verification, the sensitivity of each frame in the magnification mode was 96.1% and the sensitivity of each lesion was 100%, and a probability heat map was generated in real time to assist in lesion localization[11]. Performance comparison studies further verified the advantages of AI: a multi-center team compared the diagnostic performance of the AI system with 16-layer VGGNet and 13 endoscopic experts on 144 videos. The results showed that AI had significantly higher sensitivity in the lesion detection stage (91% vs 79%) and accuracy in the lesion characterization stage (88% vs 75%), especially for large lesions (>30mm), with a detection rate of 100%[14]. In terms of real-time auxiliary value verification, the system designed by the team had a sensitivity of 85% in detecting early ESCC in high-speed endoscopic videos (simulating the speed of conventional screening), and when assisting 18 physicians, its sensitivity increased from 45% to 52.5% ($p < 0.05$)[15]. It is worth noting that interpretability studies use CAMs to visualize the basis for AI decision-making. For example, the improved ResNet-18 model developed by the European and Asian teams showed an F1 score of 94% in 67,742 ME-NBI images. Its CAMs can accurately locate abnormal IPCLs with performance close to that of experts (F1 97%-98%) [8]. The comprehensive findings show that the AI system performs better under ME (such as a 19-29 percentage point increase in sensitivity), mainly benefiting from the objective analysis of microvascular structure. However, esophageal motility disturbances, mucosal inflammation, and anterior wall lesions under non-ME can still lead to false positives (such as misjudgment of the anatomical structure of the esophagogastric junction) or false negatives (missed diagnosis of irregular keratinized lesions). Studies have consistently confirmed that AI-assisted diagnosis can significantly shorten the diagnosis time (AI: 0.033-0.5 seconds/frame vs. expert group: 165 minutes), reduce the impact of endoscopist experience differences, and have great potential in primary care settings. Current limitations are concentrated in the bias of training data (mainly single-center) and the fact that video verification does not cover low-quality images (such as bleeding). In the future, prospective trials are needed to verify the clinical translation value and optimize the recognition accuracy of mixed IPCL patterns [11, 13, 15, 16].

## 3  Deep learning-assisted delineation of lesion margin

Small and flat lesions often exhibit subtle features. Traditional endoscopic techniques such as white light imaging and iodine staining rely on physician experience and are prone to misdiagnosis or unnecessary surgical risks. AI-assisted systems can provide real-time, objective lesion detection and precise boundary delineation, becoming a key direction for improving the efficiency and accuracy of endoscopic diagnosis[10, 17]. In a representative study, Yuan et al. developed a deep learning system based on the YOLACT model, focusing on superficial ESCC and precancerous lesions under NBI; the system adopted a multicenter retrospective and prospective three-stage design (training data 752 cases/7530 images), and verified its high performance under static images through internal and external tests (detection sensitivity 96.5%, depiction accuracy 88.9%, average intersection-over-union ratio 75.9%). In particular, through prospective clinical real-time verification (62 cancer cases), it showed a real-time diagnostic accuracy of 91.4% and a depiction accuracy of 85.9%, which was much faster than that of human physicians (12ms/image vs 21.4-33.6 seconds/image), and reached or exceeded the senior level in comparison with 11 endoscopists (including senior and junior)[17]. Another work by Yuan et al. expanded the technology to multimodal imaging integration, combining the YOLACT++ algorithm to process multiple modes such as white light imaging, NBI, magnifying endoscopy and iodine staining, aiming to detect and depict the boundaries of small (about 3 mm) flat early ESCC in real time; the system was directly integrated into the endoscopic device, and through video demonstration, it accurately captured the edge of the lesion and displayed the probability of canceration in real-time endoscopic examination, and pathological confirmation (such as invasion of the lamina propria) achieved the convenience of clinical operation without the need for additional equipment, filling the gap in AI in the multimodal real-time depiction of small

lesions[10]. Comparing these two studies, both rely on advanced instance segmentation algorithms (such as the YOLACT series) to achieve high-precision automated boundary segmentation. The results consistently highlight the potential of AI in terms of delineation accuracy (both approximately 85-90%) and real-time performance, but the focus and scope are significantly different - the NBI single-mode system is optimized for common screening needs, emphasizing the reduction of iodine staining dependence and prioritizing the verification of multi-center generalization capabilities[17], while the multi-modality system expands coverage to complex imaging combinations, focusing on the full range of feature capture of small lesions to improve clinical practicality[10]; these differences may be due to differences in research objectives (such as the former specifically evaluates the performance of ESCC and precancerous lesions in standard mode, while the latter focuses on the response of very small lesions in dynamic scenarios) and dataset composition (such as the NBI system samples contain more non-cancerous lesions to verify specificity). Comprehensive analysis shows that the automated segmentation technology of lesion boundaries can effectively reduce the endoscopist's dependence on experience through AI assistance, and has made breakthroughs in real-time detection, boundary delineation accuracy and reduction of misdiagnosis. In particular, it can significantly improve the diagnostic efficiency and the accuracy of endoscopic minimally invasive treatment planning for early ESCC, laying the foundation for its potential as a routine clinical tool[10, 17].

# 4 Deep learning-assisted assessment of tumor invasion depth and classification of lesion microvascular patterns

## 4.1 Estimation of lesion invasion depth

The prediction of the depth of invasion of esophageal squamous cell carcinoma is the core basis for the indication of endoscopic resection. The risk of lymph node metastasis of deep-layer invasion of SM2/SM3 is >25%, requiring surgical intervention[18]. The AI model based on the Japanese Endoscopic Society classification achieves accurate stratification prediction by analyzing key indicators such as the degree of destruction of microvascular configuration and tumor infiltration of submucosal glandular ducts (ductal involvement, DI). Multicenter studies have confirmed that the HRNet model has an overall accuracy rate of 80.7% in predicting deep submucosal invasion (SM2/SM3), among which the positive predictive value of B3 type blood vessels for SM2/SM3 is 100%[7, 19]. Pathological studies have revealed that DI is a characteristic sign of SM2/SM3 lesions, but DI itself does not increase the risk of metastasis (the lymph node metastasis rate in patients with mucosal carcinoma and DI is 0%). Its significance lies in reflecting the scale of tumor horizontal spread and needs to be combined with the depth of invasion assessment[20]. After training on 8660 endoscopic images, the deep neural network developed by Nakagawa achieved an accuracy of 91% in distinguishing EP-SM1 from SM2/SM3, which is equivalent to that of senior endoscopists (91%). However, the diagnostic accuracy of SM1 lesions was only 67.8%, and the main misdiagnosis was due to extramural compression artifacts[18, 21]. Shimamoto 's convolutional neural network model was verified in 102 independent videos, and the prediction accuracy of SM2/SM3 infiltration in the magnifying endoscopy mode was 89% and the specificity was 95%, which was significantly better than the average level of the expert group (accuracy of 84%)[13]. The interpretable AI system increased the physician's diagnostic sensitivity for SM2/SM3 from 37.5% to 55.4% by quantifying nine features, including avascular area size and IPCL morphology[22]. A study on the metastasis risk gradient of the submucosal layer (SM1-SM3) further validated the predictive value of AI: the metastasis rate of SM1 is approximately 22-33%, while that of SM3 is as high as 78%, supporting AI's high-risk warning for deep invasion[23].

## 4.2 Classification of lesion microvascular patterns

IPCL classification is crucial for the diagnosis and treatment of early ESCC, but its manual interpretation suffers from inter-observer variability. The AI system developed by the multi-center study is based on HRNet combined with a semantic segmentation model, which can visualize and output IPCL subtypes in real time (type A normal / low-grade intraepithelial neoplasia, type B1 high-grade intraepithelial neoplasia / lamina propria invasion, type B2 muscularis mucosa / superficial submucosal invasion, type B3 deep submucosal invasion). In 7094 ME -NBI images of 685 patients, the comprehensive accuracy of IPCL classification in the internal validation set and the external validation set reached 91.3% and 89.8%, respectively, which was significantly better than that of the

endoscopist group (78.2% in the junior group and 87.1% in the senior group)[7]. Further verification showed that the HRNet model's positioning accuracy for B1/B2/B3 vessels (intersection-over-union ratio > 0.4) supported the Japanese Endoscopic Society classification visualization and assisted junior physicians to improve the accuracy of IPCL classification by 6.5% (84.7% vs 78.2%)[7, 24]. Another study improved the Faster R-CNN architecture and integrated the polarized self-attention mechanism (PSA-HRNetV2p), achieving a detection recall rate of 79.25% for B1/B2 IPCL, including 83.94% for B1 and 74.57% for B2. The original annotation segmentation miniaturization algorithm was used to optimize small target recognition[24]. The CNN developed by Martin et al., after training on 67,742 ME-NBI images, achieved an accuracy of 91.7% in distinguishing normal from abnormal IPCL patterns, and generated CAMs in real time, revealing that the model's decision-making basis was consistent with clinical characteristics[8]. The interpretable AI system integrated multiple feature extraction models and achieved a real-time classification accuracy of 84.9% for infiltration depth in video streams. 87.5% of physicians preferred it to the traditional deep learning " black box " model[22].

## 5  Limitations and future prospects

First, most current AI-assisted diagnosis studies suffer from insufficient sample sizes. Most studies are trained on single-center, retrospective datasets. Furthermore, endoscopic images collected across different medical institutions vary in equipment and are operator-dependent, leading to biased training data[25]. While some studies have demonstrated high diagnostic accuracy, performance may decline in real-world applications across different medical institutions[25, 26]. This limitation raises questions about the generalizability of these models in real-world clinical settings[25]. Furthermore, existing studies have primarily focused on specific types of esophageal lesions, and the ability to identify early, subtle lesions and precancerous lesions requires more high-quality data[26]. Future efforts will require integrating resources from major medical institutions, establishing standardized, large-scale datasets and evaluation systems, and strengthening multicenter, prospective studies to evaluate the real-world performance of AI models[27]. Second, current AI systems are mostly single-function designs, relying heavily on single-modality data such as white-light imaging and narrow-band imaging, and lack the ability to integrate multimodal data[26, 28]. Future development should prioritize the development of multifunctional, integrated AI models that integrate functions such as lesion detection, margin delineation, and invasion depth assessment into a unified platform[29, 30]. Third, the potential negative impacts of AI on medical development must be recognized. Over-reliance on AI could lead to a decline in endoscopists' diagnostic capabilities, hinder their clinical decision-making, and affect doctor-patient communication[31–33]. Future ethical guidelines and laws and regulations will be needed to address ethical issues such as medical resource allocation, data privacy protection, algorithmic transparency, and accountability[33].

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

## Agents4Science AI Involvement Checklist

1. **Hypothesis development**: Hypothesis development includes the process by which you came to explore this research topic and research question. This can involve the background research performed by either researchers or by AI. This can also involve whether the idea was proposed by researchers or by AI.

   Answer: [A]

   Explanation: This article was proposed by researchers. Their research team has extensive experience in AI-assisted endoscopic diagnosis of early-stage esophageal squamous cell carcinoma, having previously published numerous related articles in internationally renowned medical journals. Therefore, the researchers used their developed "Lunjie" (a tool for reviewing the topic) to conduct a review, allowing for comparison with existing articles.

2. **Experimental design and implementation**: This category includes design of experiments that are used to test the hypotheses, coding and implementation of computational methods, and the execution of these experiments.

   Answer: [A]

   Explanation: Figure 1 shows the general process of AI review. It is divided into two stages: first, determining relevance, and then generating a review.

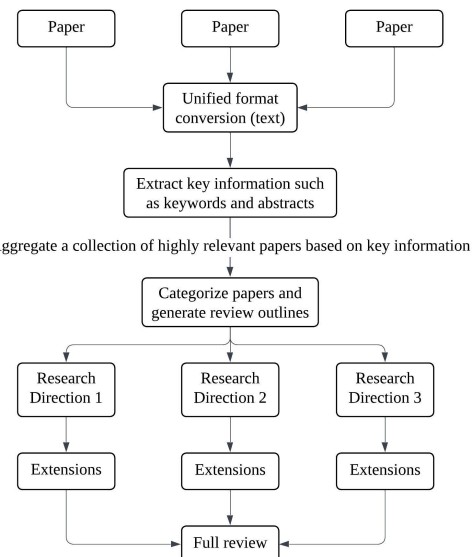

Figure 1: The general process of AI review

3. **Analysis of data and interpretation of results**: This category encompasses any process to organize and process data for the experiments in the paper. It also includes interpretations of the results of the study.

   Answer: [D]

   Explanation: This AI-generated article is a review and does not contain any experimental research. However, during the data processing process, the researchers continuously improved the model's generation quality (such as structural hierarchy and language logic) by changing the model prompt words and repeating the generation process multiple times, thus obtaining the final result.

4. **Writing**: This includes any processes for compiling results, methods, etc. into the final paper form. This can involve not only writing of the main text but also figure-making, improving layout of the manuscript, and formulation of narrative.

   Answer: [C]

   Explanation: As mentioned in Question 3, the main body of the paper was generated by AI. Since the abstract generated was unsatisfactory, the researchers instructed the AI to

regenerate it based on the main body. Furthermore, the researchers modified the language of some titles to make them more concise and understandable. Since the output was in Simplified Chinese, the researchers used Google Translate to translate it into English.

5. **Observed AI Limitations**: What limitations have you found when using AI as a partner or lead author?

Description: First, there are limitations on the selection of references. The AI developed by the research team has built-in prompt words, and automatically generates articles after selecting references, rather than the traditional method of inputting prompt words. In order to make the generated article topic focus on the target topic, it is necessary to select references with highly similar topics in advance. The researchers found that if references on other irrelevant topics are mixed in, the topic of the generated article will deviate and the ideal result will not be obtained. Secondly, to ensure that the model can operate normally, the maximum number of references cannot exceed 50; in addition, incomplete citations may occur, which may be caused by the model losing information or the model detecting that there is no writing relevance between certain references. Third, the stability of the model operation is not good, and the results generated by repeated attempts vary greatly.


# Technical Appendices and Supplementary Material

**I. Limitations (corresponding to Question 2 in Checklist 2)**

1. Core assumptions of this study (including prerequisites)

This study used AI to write the first draft based on the following three assumptions:

1.1 Assumptions on validity of literature screening:

It is assumed that AI can accurately screen out highly relevant documents based on research topics or keywords from a manually pre-processed document library. However, it should be noted that this assumption only applies to a "manually screened document library" and does not involve its ability to independently screen from massive amounts of documents and identify low-relevance documents.

1.2 Assumption of the number of literature adaptability

It is assumed that AI can normally complete review writing within a scale of 40 references. Due to the lack of large-sample testing, this hypothesis cannot be directly transferred to scenarios with a larger number of references. The stability of AI writing performance (such as topic relevance and content completeness) after the number of references increases remains to be verified.

1.3 Content authenticity assumption

It is assumed that the reference data cited by AI and the innovative content extracted from the article are authentic and reliable, without any "illusion" or fabrication. This assumption requires "manual verification after AI writing" as a supplementary verification step, through manual intervention to ensure that the cited content is consistent with the original text and the data is unbiased.

2. Potential risks and chain consequences of above hypotheses

The above assumptions may be broken due to the inherent limitations of AI technology. The specific academic risks are as follows (the risks will increase as the scale of literature expands).

2.1 The scenario where the hypothesis of the effectiveness of literature screening is broken

Without the premise of manually screening the literature library, AI-powered independent literature screening faces two major limitations: ①there is a clear upper limit to the number of articles that can be screened; ②as the total number of references increases, its adherence to the core theme decreases significantly. If the literature library is mixed with low-relevance articles, AI will have difficulty effectively screening them, which will directly cause the generated content to deviate from the intended theme and weaken the thematic focus of the review.

2.2 The scenario where the assumption of literature quantity adaptability is broken

When the number of references increases to more than 50, AI may be unable to complete the generation task due to the dual limitations of computing power and algorithms. Even if AI can complete the task, the increased data processing load will lead to a significant decline in the quality of the review writing (such as logical coherence and completeness of supporting evidence).

2.3 The scenario where the assumption of content authenticity is broken

When AI extracts key information from references, it may generate information errors. In extreme cases, it may generate "hallucination content" that is inconsistent with the original text, directly causing data errors and biased opinions in the review.

The consequences of the above scenario are progressive: first, the literature foundation of the paper is undermined (such as insufficient literature relevance and unreliable data), which in turn reduces the validity and reliability of the research conclusions, ultimately weakening the scientific nature and credibility of the entire research. Therefore, human intervention is irreplaceable in the AI-assisted writing process. Manual verification can not only verify the authenticity of the references themselves (such as their sources and core ideas), but also correct biases in AI-referenced content, thus mitigating academic risks at the source.

3. Implications based on the limitations of AI

To overcome the AI technology bottleneck exposed by the above scenarios, future models and related research can be optimized along five dimensions to enhance the applicability and reliability of AI-assisted scientific research.

3.1 Breaking through the bottleneck of literature screening capabilities

By improving the AI generation framework, adjusting core model parameters, or building a multi-agent collaborative screening mechanism, we can address the issues of "limited number of articles to be screened" and "decreasing subject adherence as the volume of literature increases", thereby enhancing its ability to handle large sample sizes and highly complex literature libraries.

3.2 Standard content generation forma

In order to reduce formatting errors in AI-generated content and improve the standardization of reviews, we can design a standardized prompt word system, clarify the unified rules for the use of professional vocabulary abbreviations and reference citation formats.

3.3 Optimize the human-machine collaboration process

The AI writing logic has been restructured, adopting a step-by-step generation model of "outline first, full text later": AI first generates a review outline that can be manually revised. After manual verification of the topic fit and structural rationality, the full text is generated based on the outline, strengthening the user's control over the content.

3.4 Close to the logic of scientific research writing

We will continue to optimize the semantic organization and argumentation logic of the model to make its expression style more consistent with the rigorous requirements of manual scientific research writing (such as the relevance of arguments and conclusions, and the sense of logical progression), reducing the sense of "machine-based expression"

3.5 Expanding multi-language application scenarios

In view of the limitation that the current model does not support English output, a multilingual generation module is introduced to cover mainstream academic languages, broaden the application scope of AI-assisted scientific research and adapt to the submission requirements of different journals.

4. Computational efficiency of the algorithm

The time required to generate a review with AI increases with the number of references. When the number is greater than 50, the model may not function properly.

5. Scope of application of research conclusions

This study used AI to generate a review, which did not include any experimental theoretical results or research conclusions. During the generation process, researchers continuously improved the model's quality (such as structural hierarchy and linguistic logic) by changing model prompts and repeating the generation process, ultimately achieving the final result. This method was only applied to approximately 35 references.

6. Privacy

The references used in the model operation process are all open access and will not cause any infringement to any individual or organization.

## II. The positive and negative impacts of AI-assisted writing (corresponding to Question 10 in Checklist 2)

1. Positive impacts

By comparing manually written reviews, the research team has developed AI that confirms the ability of generative AI to generate relatively high-quality review articles and improve writing efficiency. Simultaneously generated content can help researchers in related fields quickly understand the dynamics of the field. In addition, the application of AI in medicine has broken the boundaries of traditional disciplines, promoted interdisciplinary research cooperation, and promoted the cross-integration of medicine and AI. Most importantly, AI can help grassroots medical workers conduct scientific research.

2. Negative impacts

It cannot be denied that the use of AI may lead to the generation of false information and affect research quality, as the problem of AI-generated hallucinations has not yet been fundamentally avoided, and the generated results need to be manually verified. Additionally, researchers may overly rely on AI, reducing independent and critical thinking, affecting the originality and depth of research. The use of AI may involve academic misconduct and academic fraud. AI is open to all medical researchers, and

the possibility that some researchers will directly use the generated results as their own research results cannot be ruled out. Relevant regulations are needed to limit the improper use of AI. Because hallucinations are common in generative AI, if the generated results are directly applied to real-world practice without sufficient verification, serious consequences may result. In the future, it is necessary to establish AI-assisted checklists for authors to complete, disclose information related to AI use, or formulate standardized ethical guidelines and laws and regulations to address these issues.

# Responsible AI Statement

The AI review writing process adheres to the principles outlined in the NeurIPS Code of Ethics. We recognize the significant potential of AI for review writing and are committed to its responsible development and deployment. This statement outlines the broader impact of our work on society and the precautions we take to ensure its safe and ethical application.

## 1. Broader Impact

1.1 Positive impacts

By comparing manually written reviews, the research team has developed AI that confirms the ability of generative AI to generate relatively high-quality review articles and improve writing efficiency. Simultaneously generated content can help researchers in related fields quickly understand the dynamics of the field. In addition, the application of AI in medicine has broken the boundaries of traditional disciplines, promoted interdisciplinary research cooperation, and promoted the cross-integration of medicine and AI. Most importantly, AI can help grassroots medical workers conduct scientific research.

1.2 Negative impacts

It cannot be denied that the use of AI may lead to the generation of false information and affect research quality, as the problem of AI-generated hallucinations has not yet been fundamentally avoided, and the generated results need to be manually verified. Additionally, researchers may overly rely on AI, reducing independent and critical thinking, affecting the originality and depth of research. The use of AI may involve academic misconduct and academic fraud. AI is open to all medical researchers, and the possibility that some researchers will directly use the generated results as their own research results cannot be ruled out. Relevant regulations are needed to limit the improper use of AI. Because hallucinations are common in generative AI, if the generated results are directly applied to real-world practice without sufficient

verification, serious consequences may result. In the future, it is necessary to establish AI-assisted checklists for authors to complete, disclose information related to AI use, or formulate standardized ethical guidelines and laws and regulations to address these issues.

**2. Precautions taken to ensure the safe deployment of the AI scientist**

To mitigate the potential risks associated with the use of AI in academic writing and ensure its safe, ethical, and effective deployment, we have implemented precautions throughout our research process.

First, we clarify the scope of AI's work and its role in scholarly writing. AI's role in scholarly writing is limited to screening and synthesizing information from the literature and generating a first draft. Key tasks such as developing core ideas, critical analysis of generated results, and final conclusions remain the sole responsibility of human researchers. This clarifies the leadership of humans in scholarly writing while also preventing over-reliance on AI.

Second, AI will undergo continuous human oversight throughout the scholarly writing process. All AI-generated content, including the presentation of research facts, interpretation of experimental data, and references to literature, must undergo rigorous human verification by domain experts and original sources. This human review not only verifies the authenticity of the references themselves (such as sources and core ideas), but also corrects biases in AI-cited content, mitigating academic risks at the source.

Our deployment of AI scientists adheres to established principles of academic integrity and research ethics. We explicitly prohibit their use for any form of academic misconduct, such as plagiarism or data fabrication. The research team is committed to maintaining and adhering to evolving institutional, national, and international guidelines governing the use of AI in scholarly work.

All researchers signed security and privacy agreements; all operations involving the AI system were conducted on a secure, controlled computing platform. The literature used for training and generation consisted only of publicly available, open-access research publications to ensure no infringement of personal or institutional copyright or privacy,

and no confidential, proprietary, or personal patient data was fed into or processed by the AI.